# Predictors of exclusive breastfeeding practice among migrant and non-migrant mothers in urban China: results from a cross-sectional survey

Jia Li ,[1,2] Yifan Duan,[3] Ye Bi,[3] Jie Wang,[3] Jianqiang Lai,[3] Chen Zhao,[2] Jin Fang,[2] Zhenyu Yang[3]

¹Business School, Nanjing University of Information Science & Technology, Nanjing, China
²Child Development Center, China Development Research Foundation, Beijing, China
³National Institute for Nutrition and Health, Chinese Center for Disease Control and Prevention, Beijing, China

**Correspondence to**
Dr Zhenyu Yang;
yangzy@ninh.chinacdc.cn

## ABSTRACT

**Objective** To explore and compare the predictors for exclusive breast feeding (EBF) among migrant and non-migrant mothers in China.

**Design** A large-scale cross-sectional study.

**Setting** 12 counties/districts were covered in China.

**Participants** A total number of 10 408 mothers were recruited, of whom 3571 mothers of infants aged 0–5 months in urban China were used for analysis.

**Outcome** The practice of EBF was calculated based on the foods and drinks consumed in the last 24 hours, as recommended by WHO.

**Results** Around 30% of Chinese mothers with infants aged 0–5 months practised EBF in urban areas, with no significant difference between migrant and non-migrant mothers (p=0.433). Among the migrant mothers, factors associated with EBF included residence in big cities (adjusted OR, AOR 1.68 (95% CI 1.20 to 2.34)), premature birth (AOR 0.27 (95% CI 0.09 to 0.81)), knowledge about EBF (AOR 2.00 (95% CI 1.51 to 2.65)), low intention of breast feeding in the first month postpartum (AOR 0.59 (95% CI 0.36 to 0.97)) and mothers working in agriculture-related fields or as casual workers (AOR 1.77 (95% CI 1.18 to 2.64)). Among non-migrant mothers, in addition to similar predictors including residence in big cities (AOR 1.40 (95% CI 1.13 to 1.73)), knowledge about EBF (AOR 1.25 (95% CI 1.02 to 1.53)) and low intention of breast feeding in the first month post partum (AOR 0.46 (95% CI 0.31 to 0.70)], early initiation of breast feeding (EIBF) (AOR 1.78 (95% CI 1.35 to 2.33)) and caesarean delivery (AOR 0.74 (95% CI 0.60 to 0.89)) were also factors associated with EBF.

**Conclusions** There was no significant difference in the prevalence of EBF between migrant and non-migrant mothers in urban China. Premature birth and maternal occupation in agriculture-related fields or casual work were distinctive factors associated with EBF for migrants, while EIBF and caesarean delivery were unique predictors for non-migrants.

**Trial registration number** ChiCTR-ROC-17014148; Pre-results.

## Strengths and limitations of this study

► This is the first and largest study to explore and compare breastfeeding practices and its associated factors among migrant and non-migrant mothers in urban China.

► Mothers in large cities were more likely to practice exclusive breast feeding (EBF) when compared with mothers in medium and small cities, regardless of their migrant status.

► The mothers covered in this study may not be representative of mothers in urban areas since we only invited mothers who brought their infants to immunisation clinics to participate, which may overestimate the prevalence of EBF.

► Since it is not a nationally representative study, the results cannot be generalised to all the urban areas in China.

showed that the prevalence of breast feeding decreased with the increase in gross domestic product per capita.[1] Similar to the global trend, the prevalence of exclusive breast feeding (EBF) in China dropped from 27.6% in 2008 to 20.7% in 2013.[2 3] Rapid economic development is inseparable from an increasing number of domestic migrants.[4] China has experienced unprecedented internal migration in world history. On the one hand, better health infrastructures in cities can provide migrants with better healthcare services.[5] On the other hand, the household registration system and limited insurance coverage put migrants in a vulnerable situation with less potential access to urban benefits.[6] In addition to socioeconomic and cultural differences between migrant and non-migrant women in China, previous studies also found that migrant women had poor knowledge and limited utilisation of maternal healthcare services.[7–9]

## INTRODUCTION

China has witnessed rapid economic growth in the past decades. Cross-country evidence

A large number of previous studies have focused on immigrant mothers, who moved to a country that was not their country of origin.[10] Immigrant mothers were found to be more likely to initiate and continue breast feeding than non-immigrant mothers. However, EBF remained a challenge for both groups.[10] Little attention has been paid to breastfeeding practices and its associated factors for domestic migrant mothers. Previous literature in China disclosed the disparities in breastfeeding practices between urban and rural areas,[3 11] employed and unemployed mothers,[12] Han and other minority groups,[13] and highly educated and less-educated mothers.[14] A wide range of individual, sociodemographic, cultural, psychosocial and environmental factors had been identified as risk factors of EBF in China,[15 16] such as maternal and child characteristics,[12 13 17–19] support from family members and friends,[18 20 21] maternity facility education, support and practice[11 17 19 22] as well as breastfeeding intention.[18] Yet, few studies have examined the difference in breastfeeding practice and its predictors among migrant and non-migrant mothers in China. To fill this knowledge gap, we analysed large-scale cross-sectional survey data to examine the similarities and disparities in EBF and its associated factors, among migrant and non-migrant mothers in China.

## METHODS
### Study design and participants
Data for this study were derived from a large-scale cross-sectional survey in China initiated by the China Development Research Foundation in 2017. This survey was designed by an independent team at the National Institute for Nutrition and Health of the Chinese Center for Disease Control and Prevention (NINH, China CDC). Calculation of sample size was based on the expected prevalence of EBF for infants aged 0–5 months.

We adopted a multistage stratified cluster sampling approach while selecting the survey sample. In the first stage, all districts/counties were categorised into four strata, of which 12 districts/counties were selected (four from large cities, four from medium and small cities, two from normal rural areas and two from poor rural areas). The selection of districts/counties considered their geographic distributions, executive capacities, collaboration of the provincial level CDC and their population sizes. In the second stage, four clusters were randomly selected using the probability proportional to size sampling in each selected district/county. One cluster usually had one community health centre. In the last stage, mothers who brought their infants aged 0–11 months to immunisation clinics were randomly invited to participate in the survey until the designed sample size was met. The inclusion criteria for this study were mothers of infants aged 0–11 months who signed the informed consent form, had no psychiatric disorders, and were able to answer questions clearly. To decrease the reporting bias of feeding practice, we only invited mothers who were the primary caregivers of their infants. In total, 10408 mothers of infants aged 0–11 months were interviewed.

### Patient and public involvement
This survey was undertaken by mothers of infants aged 0–11 months in China to understand the prevalence of EBF and its associated factors. Neither these mothers nor the public were involved in the study design and implementation. The study results will be disseminated to the public through media briefings and scientific publications.

### Data collection
This survey was conducted by the NINH, China CDC in collaboration with 12 teams at the provincial level CDC from September 2017 to January 2018. Data were collected through face-to-face interviews using a structured questionnaire programmed into smartphones or tablets. The questionnaire was developed by NINH, China CDC, which comprised eight sections: demographic characteristics, breastfeeding practices, maternal and child health, supportive environment for breast feeding, health service, workplace, social environment and culture, and household financial situation-related factors. We obtained written consent forms from all mothers.

### Statistical analysis
We used the past 24 hours infant and young child feeding indicator method recommended by WHO to assess the feeding practice of infants aged 0–5 months to generate internationally comparable rates of EBF. We defined the practice of being fed exclusively with breast milk in the last 24 hours as EBF. Even though previous studies indicated that providing prelacteal feeds is a long-held tradition in many parts of China, regional disparities were quite large.[23] Additionally, water is the top one first drink received by the newborns in China and sometimes mothers revert to EBF after breast milk came in.[24] Using this method can reduce the recall bias of mothers with older infants as well as the possibility of underestimating EBF.[24]

We first used descriptive analysis to report the selected characteristics, and then assessed their differences between migrant and non-migrant mothers, using Pearson's $X^2$ test. Potential predictors were selected through reviewing relevant literature.[3 12–15 18 25] $X^2$ tests were then used to examine the potential predictors of EBF for migrant and non-migrant mothers, respectively.

Sociodemographic characteristics included maternal age (≤25 years old, 26–35 years old and ≥36 years old), maternal ethnicity (Han and others), maternal and paternal education (junior high school or lower, high school and college or higher education), maternal and paternal occupation (unemployed, agriculture-related or casual work, industry or business related, professionals or white-collars), employment type of mothers (formal and informal) and place of residence (big cities, and small and medium cities). Obstetrical history comprised parity

(1 and≥2), premature birth (yes and no), low birth weight (yes and no), mode of delivery (vaginal and caesarean) and delivery hospital type (hospitals at the municipal level or above level, hospitals at the district/county level and others). Breastfeeding-related factors included early initiation of breast feeding (EIBF) (yes and no), knowledge about EBF (yes and no), low intention of breast feeding within 1 month post partum (yes and no), and postpartum breastfeeding difficulties (yes and no). We defined EIBF as the practice of putting infants to the breast of their mothers within the first hour of birth based on the definition of the WHO.[26] We defined mothers who were always, very often or sometimes unwilling to breast-feed within 1 month post partum as having a low intention of breast feeding within 1 month post partum. Social influence denoted whether her partner, friend, mother or mother-in-law agreed that breastmilk was better than breastmilk substitutes. Health-seeking behaviours comprised the timing of considering how to feed the infants and attending antenatal visits, breastfeeding education sessions or mother groups.

We then performed logistic regressions to identify predictors of EBF. The selection of covariate variables was based on the p values of $X^2$ tests ($p < 0.05$). ORs and 95% CIs were displayed. The data analyses were performed by Stata V.15.0 (Stata). We set the level of statistical significance at $\alpha = 0.05$.

## RESULTS

### Sample description and factors associated with EBF

Considering that migrants usually concentrate in urban areas, we only focused on 3571 mothers of infants aged 0–5 months who lived in big, medium and small cities. Migrant mothers were defined as those who had resided in a specific county for a month or more and had been in a place different from their registered county. We excluded 172 observations due to the missing values of potential predictors, which left 3399 (>95% of the original sample) observations for statistical analysis, of which 2199 were non-migrants and 1200 were migrants. A sensitivity analysis using the full sample revealed a very similar migrant distribution to the one shown in this study.

Table 1 displays the summary statistics and results of the bivariate analysis, reflecting significant differences in the distribution of selected characteristics and associated factors of EBF among non-migrant and migrant mothers. The prevalence of EBF was 30.3% among non-migrant mothers, which was not statistically different from 29.0% among migrant mothers (p=0.433).

Migrant mothers were statistically different from non-migrant mothers in age, ethnicity, occupation (her and her partner's), formal employment status and place of residence. More than 60% of mothers were aged 26–35 years in both groups, while the proportion of mothers above 36 years was significantly higher in non-migrant mothers (p<0.001). The likelihood of being of Han ethnicity, working as a professional or a white-collar worker, and

being formally employed was statistically higher in non-migrant mothers than migrant mothers as well (table 1). Partners of non-migrant mothers were more likely to work as professionals or white-collar workers. For non-migrant mothers, sociodemographic characteristics like maternal education and occupation, paternal education, employment status of mothers and place of residence were significantly associated with EBF (p<0.001). For migrant mothers, in addition to maternal education and occupation, paternal education, employment status of mothers and place of residence, the ethnicity of mothers also significantly affected the prevalence of EBF. Mothers or their partners with a college or higher education were more likely to practice EBF in both groups. The prevalence of EBF was higher in mothers who had a job, were formally employed, and lived in big cities among both migrant and non-migrant mothers. Mothers from the Han ethnic group were more likely to practice EBF than mothers from minority groups among the migrant group (table 1).

Among factors about obstetrical history, the percentage of having only one child was statistically higher among migrant mothers than among non-migrant mothers (p<0.001). The proportions of premature birth and low birth weight were quite similar between migrant and non-migrant mothers, while the rate of caesarean section was statistically higher in non-migrant mothers than in migrant mothers. Caesarean section was significantly associated with a lower prevalence of EBF for both migrant and non-migrant mothers. However, for migrant women, premature birth and low birth weight of infants were also negatively related to EBF. A significantly larger proportion of migrant mothers delivered their infants in hospitals at the municipal level or above (p<0.001). Mothers who delivered infants in hospitals at the district/county level were less likely to practice breast feeding in both groups.

EIBF, having a low intention of breast feeding within 1 month post partum and breastfeeding difficulties postpartum, occurred more frequently among migrant mothers than among non-migrant mothers, while only the prevalence of having breastfeeding difficulties post partum was statistically different between the two groups. Unlike the non-migrant group where EIBF was associated with EBF significantly (p<0.001), migrant mothers were more likely to practice EBF if they had breastfeeding difficulties postpartum. More than half of the mothers knew the definition of EBF, and around 10% of them had a low intention of breastfeeding within 1 month post partum in both groups. Mothers in both groups were more likely to practise EBF if they knew its definition and were less likely to do it if their intentions of breast feeding were low within 1 month post partum.

Even though more than 80% of mothers received positive influence from their partners, friends, mothers or mothers-in-law about breastmilk being better than breastmilk substitutes, these influences differed among migrant and non-migrant mothers. The influence from partners and friends was significantly associated with EBF

 

**Table 1** Factors associated with EBF practice among mothers of infants aged 0–5 months by migration status

| | Non-migrant mothers (n=2199) | | | | | Migrant mothers (n=1200) | | | | | | |
| --- | --- | --- | --- | --- | --- | --- | --- | --- | --- | --- | --- | --- |
| | Total | | EBF | | P value* | Total | | EBF | | P value* | P value† |
| | N | % | n | % | | N | % | n | % | | |
| **Sociodemographic characteristics** | | | | | | | | | | | |
| Maternal age | | | | | 0.740 | | | | | 0.536 | <0.001 |
| ≤25 years old | 448 | 20.4 | 129 | 28.8 | | 283 | 23.6 | 83 | 29.3 | | |
| 26–35 years old | 1411 | 64.2 | 432 | 30.6 | | 799 | 66.6 | 236 | 29.5 | | |
| 36+ years old | 340 | 15.5 | 105 | 30.9 | | 118 | 9.8 | 29 | 24.6 | | |
| Maternal ethnicity | | | | | 0.080 | | | | | 0.019 | 0.002 |
| Others | 360 | 16.4 | 123 | 34.2 | | 248 | 20.7 | 57 | 23.0 | | |
| Han | 1839 | 83.6 | 543 | 29.5 | | 952 | 79.3 | 291 | 30.6 | | |
| Maternal education | | | | | <0.001 | | | | | <0.001 | 0.489 |
| Junior high school or lower | 666 | 30.3 | 167 | 25.1 | | 361 | 30.1 | 79 | 21.9 | | |
| High school | 422 | 19.2 | 106 | 25.1 | | 212 | 17.7 | 52 | 24.5 | | |
| College or higher education | 1111 | 50.5 | 393 | 35.4 | | 627 | 52.3 | 217 | 34.6 | | |
| Maternal occupation | | | | | <0.001 | | | | | 0.001 | <0.001 |
| Unemployed | 776 | 35.3 | 201 | 25.9 | | 449 | 37.4 | 101 | 22.5 | | |
| Agriculture-related or casual work | 370 | 16.8 | 101 | 27.3 | | 211 | 17.6 | 70 | 33.2 | | |
| Industry or business-related | 325 | 14.8 | 106 | 32.6 | | 237 | 19.8 | 82 | 34.6 | | |
| Professionals or white-collars | 728 | 33.1 | 258 | 35.4 | | 303 | 25.3 | 95 | 31.4 | | |
| Paternal education | | | | | <0.001 | | | | | <0.001 | 0.211 |
| Junior high school or lower | 696 | 31.7 | 174 | 25.0 | | 345 | 28.8 | 80 | 23.2 | | |
| High school | 404 | 18.4 | 90 | 22.3 | | 233 | 19.4 | 54 | 23.2 | | |
| College or higher education | 1099 | 50.0 | 402 | 36.6 | | 622 | 51.8 | 214 | 34.4 | | |
| Paternal occupation | | | | | 0.434 | | | | | 0.473 | <0.001 |
| Unemployed | 112 | 5.1 | 37 | 33.0 | | 49 | 4.1 | 11 | 22.4 | | |
| Agriculture-related or casual work | 597 | 27.1 | 169 | 28.3 | | 317 | 26.4 | 101 | 31.9 | | |
| Industry or business-related | 660 | 30.0 | 195 | 29.5 | | 453 | 37.8 | 127 | 28.0 | | |
| Professionals or white-collars | 830 | 37.7 | 265 | 31.9 | | 381 | 31.8 | 109 | 28.6 | | |
| Mother was formally employed | | | | | <0.001 | | | | | <0.001 | 0.021 |
| No | 1356 | 61.7 | 365 | 26.9 | | 788 | 65.7 | 197 | 25.0 | | |
| Yes | 843 | 38.3 | 301 | 35.7 | | 412 | 34.3 | 151 | 36.7 | | |
| Place of residence | | | | | <0.001 | | | | | <0.001 | <0.001 |
| Small and medium-sized cities | 1222 | 55.6 | 299 | 24.5 | | 411 | 34.3 | 82 | 20.0 | | |
| Big cities | 977 | 44.4 | 367 | 37.6 | | 789 | 65.8 | 266 | 33.7 | | |

Continued

**Table 1** Continued

| | Non-migrant mothers (n=2199) | | | | | Migrant mothers (n=1200) | | | | | |
| --- | --- | --- | --- | --- | --- | --- | --- | --- | --- | --- | --- |
| | Total | | EBF | | P value* | Total | | EBF | | P value* | P value† |
| | N | % | n | % | | N | % | n | % | | |
| **Obstetrical history** | | | | | | | | | | | |
| Parity | | | | | 0.875 | | | | | 0.105 | <0.001 |
| ≥2 | 1181 | 53.7 | 356 | 30.1 | | 509 | 42.4 | 135 | 26.5 | | |
| 1 | 1018 | 46.3 | 310 | 30.5 | | 691 | 57.6 | 213 | 30.8 | | |
| Premature birth | | | | | 0.877 | | | | | 0.006 | 0.430 |
| No | 2112 | 96.0 | 639 | 30.3 | | 1159 | 96.6 | 344 | 29.7 | | |
| Yes | 87 | 4.0 | 27 | 31.0 | | 41 | 3.4 | 4 | 9.8 | | |
| Low birth weight | | | | | 0.389 | | | | | 0.019 | 0.807 |
| No | 2104 | 95.7 | 641 | 30.5 | | 1146 | 95.5 | 340 | 29.7 | | |
| Yes | 95 | 4.3 | 25 | 26.3 | | 54 | 4.5 | 8 | 14.8 | | |
| Mode of delivery | | | | | <0.001 | | | | | 0.016 | 0.010 |
| Vaginal | 1329 | 60.4 | 438 | 33.0 | | 779 | 64.9 | 244 | 31.3 | | |
| Caesarean | 870 | 39.6 | 228 | 26.2 | | 421 | 35.1 | 104 | 24.7 | | |
| Delivery hospital type | | | | | 0.103 | | | | | 0.095 | <0.001 |
| Hospitals at the municipal or above level | 1348 | 61.3 | 423 | 31.4 | | 849 | 70.8 | 259 | 30.5 | | |
| Hospitals at the district/county level | 796 | 36.2 | 222 | 27.9 | | 316 | 26.3 | 77 | 24.4 | | |
| Others | 55 | 2.5 | 21 | 38.2 | | 35 | 2.9 | 12 | 34.3 | | |
| **Breastfeeding-related factors** | | | | | | | | | | | |
| Early initiation of breast feeding | | | | | <0.001 | | | | | 0.098 | 0.216 |
| No | 1917 | 87.2 | 538 | 28.1 | | 1028 | 85.7 | 289 | 28.1 | | |
| Yes | 282 | 12.8 | 128 | 45.4 | | 172 | 14.3 | 59 | 34.3 | | |
| Knowledge about EBF | | | | | <0.001 | | | | | <0.001 | 0.708 |
| No | 940 | 42.7 | 238 | 25.3 | | 505 | 42.1 | 98 | 19.4 | | |
| Yes | 1259 | 57.3 | 428 | 34.0 | | 695 | 57.9 | 250 | 36.0 | | |
| Low intention of breast feeding within 1 month post partum | | | | | <0.001 | | | | | 0.029 | 0.115 |
| No | 2018 | 91.8 | 635 | 31.5 | | 1082 | 90.2 | 324 | 29.9 | | |
| Yes | 181 | 8.2 | 31 | 17.1 | | 118 | 9.8 | 24 | 20.3 | | |
| Breastfeeding difficulties post partum | | | | | 0.436 | | | | | <0.001 | 0.022 |
| No | 1567 | 71.3 | 467 | 29.8 | | 810 | 67.5 | 210 | 25.9 | | |
| Yes | 632 | 28.7 | 199 | 31.5 | | 390 | 32.5 | 138 | 35.4 | | |
| **Social influence** | | | | | | | | | | | |
| Partner supports breast feeding | | | | | 0.012 | | | | | 0.031 | 0.868 |

Continued

**Table 1** Continued

| | Non-migrant mothers (n=2199) | | | | | Migrant mothers (n=1200) | | | | | |
|---|---|---|---|---|---|---|---|---|---|---|---|
| | **Total** | | **EBF** | | | **Total** | | **EBF** | | | |
| | N | % | n | % | P value* | N | % | n | % | P value* | P value† |
| No | 270 | 12.3 | 64 | 23.7 | | 145 | 12.1 | 31 | 21.4 | | |
| Yes | 1929 | 87.7 | 602 | 31.2 | | 1055 | 87.9 | 317 | 30.0 | | |
| **Friend supports breast feeding** | | | | | | | | | | | |
| No | 318 | 14.5 | 80 | 25.2 | 0.031 | 179 | 14.9 | 45 | 25.1 | 0.217 | 0.719 |
| Yes | 1881 | 85.5 | 586 | 31.2 | | 1021 | 85.1 | 303 | 29.7 | | |
| **Mother supports breast feeding** | | | | | | | | | | | |
| No | 255 | 11.6 | 70 | 27.5 | 0.295 | 155 | 12.9 | 32 | 20.6 | 0.014 | 0.259 |
| Yes | 1944 | 88.4 | 596 | 30.7 | | 1045 | 87.1 | 316 | 30.2 | | |
| **Mother-in-law supports breast feeding** | | | | | | | | | | | |
| No | 312 | 14.2 | 92 | 29.5 | 0.740 | 147 | 12.3 | 36 | 24.5 | 0.198 | 0.114 |
| Yes | 1887 | 85.8 | 574 | 30.4 | | 1053 | 87.8 | 312 | 29.6 | | |
| **Health-seeking behaviours** | | | | | | | | | | | |
| **Timing of considering how to feed the infants** | | | | | | | | | | | |
| Before pregnancy | 1208 | 54.9 | 365 | 30.2 | 0.490 | 532 | 44.3 | 178 | 33.5 | 0.002 | <0.001 |
| During pregnancy | 646 | 29.4 | 188 | 29.1 | | 429 | 35.8 | 119 | 27.7 | | |
| After birth | 345 | 15.7 | 113 | 32.8 | | 239 | 19.9 | 51 | 21.3 | | |
| **Attend antenatal visits** | | | | | | | | | | | |
| No | 89 | 4.0 | 27 | 30.3 | 0.992 | 65 | 5.4 | 23 | 35.4 | 0.243 | 0.067 |
| Yes | 2110 | 96.0 | 639 | 30.3 | | 1135 | 94.6 | 325 | 28.6 | | |
| **Attend breastfeeding education sessions** | | | | | | | | | | | |
| No | 1101 | 50.1 | 312 | 28.3 | 0.046 | 614 | 51.2 | 169 | 27.5 | 0.249 | 0.540 |
| Yes | 1098 | 49.9 | 354 | 32.2 | | 586 | 48.8 | 179 | 30.5 | | |
| **Attend mother groups** | | | | | | | | | | | |
| No | 1373 | 62.4 | 434 | 31.6 | 0.082 | 769 | 64.1 | 223 | 29.0 | 0.999 | 0.342 |
| Yes | 826 | 37.6 | 232 | 28.1 | | 431 | 35.9 | 125 | 29.0 | | |

* P value of X² test to identify risk factors of exclusive breastfeeding practice among migrant and non-migrant mothers, respectively.
†P value of X² test to compare distribution of covariate variables between migrant and non-migrant mothers.
EBF, exclusive breast feeding.

for non-migrant mothers, while the influence from their partners and their own mothers were significantly associated with EBF for migrant mothers.

Non-migrant mothers were more likely to adopt health-seeking behaviours like considering how to feed their infants before pregnancy and attending antenatal visits, breastfeeding education sessions and mother groups, than migrant mothers. Only the timing of considering how to feed their infants was statistically different between the two groups. Migrant mothers who considered how to feed their infants before pregnancy were more likely to practice EBF. Attending breastfeeding education sessions significantly affected the EBF practice for non-migrant mothers only.

## Predictors of breastfeeding practice among migrant and non-migrant mothers

We then performed logistic regression to determine the predictors of EBF, respectively. We only kept covariates with p values less than 0.05 in table 1.

Among sociodemographic factors, place of residence was significantly associated with EBF. Migrants living in big cities were more likely to practise EBF than those in small or medium cities (adjusted OR (AOR)=1.68, 95% CI 1.20 to 2.34) after adjusting for other potential confounders. The odds of EBF were lower among migrant mothers with preterm infants (AOR=0.27, 95% CI 0.09 to 0.81) and having a low intention of breastfeeding in the first month post partum (AOR=0.59, 95% CI 0.36 to 0.97). Migrant mothers who worked in agriculture-related fields or as casual workers (AOR=1.77, 95% CI 1.18 to 2.64), and who knew EBF (AOR=2.00, 95% CI 1.51 to 2.65) were more likely to practise EBF (table 2).

Among non-migrant mothers, we found that residence in big cities (AOR=1.40, 95% CI 1.13 to 1.73), practising EIBF (AOR=1.78, 95% CI 1.35 to 2.33) and knowledge about EBF (AOR=1.25, 95% CI 1.02 to 1.53) were significantly associated with optimal breastfeeding practice. However, caesarean section (AOR=0.74, 95% CI 0.60 to 0.89) and low intention of breast feeding (AOR=0.46, 95% CI 0.31 to 0.70) significantly decreased the likelihood to practise EBF (table 3).

## DISCUSSION

In this study, we found that breastfeeding practice was suboptimal in both migrant and non-migrant mothers. The prevalence of EBF among non-migrant women was not significantly different from that of migrant mothers. This is the first and largest study to explore and compare breastfeeding practices and their potential predictors among migrant and non-migrant mothers in China. Multivariate regression results suggested that similar predictors of EBF for mothers in both groups included residence in big cities, knowledge about EBF and low intention of breast feeding in the first month post partum. Mothers in big cities were approximately 1.5 times more likely to practise EBF. Among migrant mothers, having premature

**Table 2** Predictors of EBF practice among migrant mothers of infants aged 0–5 months (n=1200)

| Risk factors | AOR | 95% CI |
|---|---|---|
| Maternal ethnicity | | |
| Others | 1 | |
| Han | 1.14 | 0.79 to 1.65 |
| Maternal education | | |
| Junior high school or lower | 1 | |
| High school | 1.15 | 0.72 to 1.83 |
| College or higher education | 1.38 | 0.83 to 2.28 |
| Maternal occupation | | |
| Unemployed | 1 | |
| Agriculture-related or casual work | 1.77** | 1.18 to 2.64 |
| Industry or business-related | 1.25 | 0.84 to 1.86 |
| Professionals or white-collars | 0.98 | 0.61 to 1.55 |
| Paternal education | | |
| Junior high school or lower | 1 | |
| High school | 0.71 | 0.45 to 1.13 |
| College or higher education | 0.90 | 0.55 to 1.47 |
| Mother was formally employed | | |
| No | 1 | |
| Yes | 1.29 | 0.88 to 1.91 |
| Place of residence | | |
| Small and medium cities | 1 | |
| Big cities | 1.68** | 1.20 to 2.34 |
| Premature birth | | |
| No | 1 | |
| Yes | 0.27* | 0.09 to 0.81 |
| Low birth weight | | |
| No | 1 | |
| Yes | 0.66 | 0.29 to 1.52 |
| Mode of delivery | | |
| Vaginal | 1 | |
| Caesarean | 0.86 | 0.65 to 1.14 |
| Knowledge about EBF | | |
| No | 1 | |
| Yes | 2.00*** | 1.51 to 2.65 |
| Low intention of breastfeeding within 1 month post partum | | |
| No | 1 | |
| Yes | 0.59* | 0.36 to 0.97 |
| Breastfeeding difficulties post partum | | |
| No | 1 | |
| Yes | 1.32 | 0.99 to 1.75 |
| Partner supports breast feeding | | |
| No | 1 | |
| Yes | 1.34 | 0.77 to 2.35 |
| Grandmother supports breast feeding | | |
| No | 1 | |

Continued

**Table 2** Continued

| Risk factors | AOR | 95% CI |
|---|---|---|
| Yes | 1.51 | 0.88 to 2.60 |
| Timing of considering how to feed the infants | | |
| Before pregnancy | 1 | |
| During pregnancy | 0.87 | 0.65 to 1.17 |
| After birth | 0.75 | 0.51 to 1.11 |

ORs were adjusted for all the covariates in the model.
*P<0.05, **P<0.01, ***P<0.001.
AOR, Adjusted OR; EBF, exclusive breast feeding.

**Table 3** Predictors of EBF practice among non-migrant mothers of infants aged 0–5 months (n=2199)

| Risk factors | AOR | 95% CI |
|---|---|---|
| Maternal education | | |
| Junior high school or lower | 1 | |
| High school | 0.98 | 0.70 to 1.36 |
| College or higher education | 0.98 | 0.66 to 1.45 |
| Maternal occupation | | |
| Unemployed | 1 | |
| Agriculture-related or casual work | 1.15 | 0.85 to 1.54 |
| Industry or business-related | 1.15 | 0.84 to 1.59 |
| Professionals or white-collar | 1.20 | 0.85 to 1.70 |
| Paternal education | | |
| Junior high school or lower | 1 | |
| High school | 0.81 | 0.58 to 1.13 |
| College or higher education | 1.33 | 0.92 to 1.93 |
| Mother was formally employed | | |
| No | 1 | |
| Yes | 0.98 | 0.72 to 1.35 |
| Place of residence | | |
| Small and medium cities | 1 | |
| Big cities | 1.40** | 1.13 to 1.73 |
| Mode of delivery | | |
| Vaginal | 1 | |
| Caesarean | 0.74** | 0.60 to 0.89 |
| Early initiation of breast feeding | | |
| No | 1 | |
| Yes | 1.78*** | 1.35 to 2.33 |
| Knowledge about EBF | | |
| No | 1 | |
| Yes | 1.25* | 1.02 to 1.53 |
| Low intention of breast feeding within 1 month post partum | | |
| No | 1 | |
| Yes | 0.46*** | 0.31 to 0.70 |
| Partner supports breast feeding | | |
| No | 1 | |
| Yes | 1.22 | 0.84 to 1.78 |
| Friend supports breast feeding | | |
| No | 1 | |
| Yes | 1.14 | 0.81 to 1.61 |
| Attend breastfeeding education sessions | | |
| No | 1 | |
| Yes | 1.11 | 0.92 to 1.36 |

ORs were adjusted for all the covariates in the model.
*P<0.05, **P<0.01, ***P<0.001.
AOR, adjusted OR; EBF, exclusive breast feeding.

infants and working in agriculture-related fields or as casual workers, were associated with EBF practice. However, for non-migrant mothers, EIBF and caesarean delivery were unique predictors of EBF practice.

Living in big cities was the only common variable in demographic characteristics that were significantly associated with EBF practice in both groups. Previous literature also suggested a higher rate of EBF in big cities than in middle and small cities in China using nationally representative data derived from the Chinese National Nutrition and Health Survey conducted in 2013.[3] This is possibly due to better knowledge and a supportive environment for mothers in big cities than for those living in small and medium cities regardless of their migration status. For example, the proportion of mothers who received information on encouraging breastfeeding was higher in big cities (55.3%) than in small and medium cities (44.7%), and more than 92% of them received encouragement from the delivery hospital in big cities. This revealed that both migrant and non-migrant mothers may benefit from better breastfeeding-related education and support, in big cities in China.

Another common predictor of EBF for both migrant and non-migrant mothers was knowledge about EBF. Some of the previous studies used knowledge scores on breastfeeding benefits as a predictor of breastfeeding practice, but they failed to provide consistent evidence on its association with EBF.[11 18] However, we only focused on the key message that infants should be exclusively breastfed for the first 6 months. This implies that future education intervention projects should pay more attention to disseminating core messages in promoting optimal breastfeeding practices.

Additionally, we also found that having a low intention of breast feeding in the first month post partum significantly decreased the odds of practising EBF for both groups. Cracked nipples or nipple pain, and insufficient breastmilk supply ranked as the top two reasons, which together accounted for around half of all the answers causing low breastfeeding intention. In alignment with previous studies, nipple pain or fissure within the first month postpartum was the most common reason for the low intention of breastfeeding among mothers regardless of their migration status.[27 28] Given that nipple

pain was often attributed to incorrect positioning and attachment,[28] healthcare providers need to find effective measures to help establish proper positioning for

mothers, good attachment of the babies to the breast as well as effective sucking. In addition, insufficient milk supply was frequently reported by previous studies as the leading cause of breastfeeding cessation or its exclusivity.[29][30] However, there is limited evidence on how to design interventions to address maternal perceptions or concerns of insufficient milk supply.[31][32] Further studies are needed to identify effective interventions regarding the maternal perception of milk supply.

In addition to the common predictors of EBF, as explained above, there were unique risk factors for migrant and non-migrant mothers, respectively. Among migrant mothers, having premature infants significantly decreased the prevalence of EBF. The EBF rate for migrant mothers with preterm infants was 9.8%, which was vastly lower than that of non-migrant mothers (31.0%). The low EBF rate could be attributable to mother–infant separation in neonatal intensive care units (NICUs) in China,[33] which resulted in many difficulties and challenges for mothers with premature infants. The disparity in the EBF rate of premature infants may come from the different choices of hospitalisation services between migrant and non-migrant mothers. Even though migrant mothers are more likely to deliver their infants in higher level hospitals in general, the proportion of migrant mothers with premature infants who delivered in hospitals at the municipal or higher levels was 65.9%, lower than that of non-migrant mothers (73.6%). In China, high-level hospitals are believed to have well-trained health workers and high-quality medical care.[34] An internationally well-known practice known as kangaroo care (KC) has only been implemented in some NICUs of high-level hospitals in China as pilot studies, for promoting the development of premature infants.[35] Implementation of KC was hindered by the limits on parental visitation, inconsistent guidelines, and safety fears of the medical staff.[35] Medical practice in NICUs may need to encourage more involvement of parents to care for their infants.

Additionally, migrant mothers working in agriculture-related fields or as casual workers were more likely to practise EBF. This is possibly because when compared with mothers working in other fields, they have more flexible time, and the shortest commuting time of 29 min for those who have already gone back to work. When compared with unemployed mothers, these migrant mothers were more likely to adopt health-seeking behaviours like starting to consider how to feed their infants before pregnancy and attending breastfeeding education sessions in hospitals. These findings were in alignment with previous studies, finding that mothers who worked full time were less likely to initiate or continue breast feeding.[36] Our results suggested that working was not necessarily a barrier to breastfeeding. Allowing lactating mothers to work with a flexible schedule that may provide support for breast feeding.

Among non-migrant mothers, EIBF and caesarean section were distinctive predictors of EBF. The rate of EIBF was 12.8% in non-migrant mothers, which was slightly higher than the 9% found in a study in the Sichuan province,[37] but much lower than the national prevalence of 28.7% in 2013[38] and the 59.4% found in a study in central and western China.[39] A large regional disparity of EIBF in China may be due to the delayed process of implementing Early Essential Newborn Care (EENC) in China. Even in places where EENC had been implemented, skin-to-skin contact was often interrupted due to inadequate facilities and EIBF was difficult to be ensured.[40] Hospitals in China need to speed up their process of adopting EENC recommended practices and national policies are needed to be established to help hospitals and health workers overcome obstacles in implementing these practices.

For obstetrical history-related factors, we also found that the prevalence of caesarean delivery was high and negatively associated with EBF among non-migrant mothers only. The prevalence of caesarean delivery was 39.6% among non-migrant mothers, which was higher than the national average of 36.7% in 2018.[41] Even though China had made efforts in decreasing caesarean delivery in the past decade,[42] the national prevalence is still far higher than the suggested prevalence of 10%–15% by the WHO.[43] The government and health facilities need to make effective policies and measures in addressing non-clinical reasons for preferring a caesarean delivery, such as maternal request and perceived convenience.[44]

This study derived data from a large-scale cross-sectional survey covering eight sample sites in urban areas, which is the first and largest study to explore and compare breast-feeding practices and their potential predictors among migrant and non-migrant mothers in China. Findings in this study filled the knowledge gap in the similarities and differences in the predictors of EBF among migrant and non-migrant mothers, which had important implications for future studies as well as health interventions to promote optimal breastfeeding practices in China.

However, our study still faced the following limitations. First, differences in the predictors of EBF among migrant and non-migrant mothers revealed in this study, may come from different choices in hospitalisation services. Migrant mothers may have restricted access to better health services due to social exclusions, lack of social networks and health insurance. Future studies need to take a closer look at the differences in health facility choices among migrant and non-migrant mothers and identify potential obstacles faced by the migrants. Second, mothers covered in this study may not be representative of the mothers in general. That is because we only interviewed mothers who brought their infants into immunisation clinics at the time of the interview, which naturally excluded children who were left behind in their early childhood and may overestimate the prevalence of EBF for both groups.[45][46] A low rate of EBF would be really problematic since it will not only result in nutrition-related harm to the early childhood development of children, but also increase the health risks of mothers in China.[1] Third, when selecting sample sites in the first stage, we considered the executive capacities, and collaboration of the provincial level

## Open access

CDC. Thus, sample sites selected in this study to some extent reflected a higher efficiency of health systems at the county/district level. Caution is needed when interpreting these results and the results disclosed in this study cannot be generalised to the whole urban areas of China. Fourth, responses of the mothers may be influenced by social desirability bias since data collection occurred in community health centres, where mothers usually received education on EBF.

To conclude, this study found that there was no significant difference in the prevalence of EBF between migrant and non-migrant mothers in our sampled urban areas of China. Even though the predictors of EBF shared some similarities between migrant and non-migrant mothers, future interventions may still need to adopt different strategies in promoting optimal breastfeeding practices in different groups.

**Acknowledgements** The authors would like to thank research teams from 12 sample sites for their hard work in the data collection. The authors also want to thank all the mothers who participated in this study.

**Contributors** JLi, YD, ZY and JLai designed the study; YD, YB, JW and ZY supervised the field study; JLi, YD and ZY acquired and analysed the data; JLi interpreted the data and drafted the first manuscript; and ZY, YD, CZ and JF provided critical intellectual feedback to help revise the manuscript. All authors have read and approved the final manuscript.

**Funding** The study was funded by the Bill & Melinda Gates Foundation to China Development Research Foundation (grant number OPP1152715).

**Disclaimer** The views expressed in the article are those of the authors and do not necessarily reflect the views of the institution and funder.

**Competing interests** None declared.

**Patient consent for publication** Not required.

**Ethics approval** The study design was approved by the Medical Ethics Committee at the NINH, China CDC.

**Provenance and peer review** Not commissioned; externally peer reviewed.

**Data availability statement** Data are available on reasonable request.

**ORCID iD**
Jia Li http://orcid.org/0000-0002-9798-9892

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
