## [Reviewer comments · BMJ Open]

ARTICLE DETAILS

TITLE (PROVISIONAL)	Predictors of exclusive breastfeeding practice among migrant and non-migrant mothers in urban China: results from a cross-sectional survey
AUTHORS	Li, Jia; Duan, Yifan; Bi, Ye; Wang, Jie; Lai, Jianqiang; Zhao, Chen; Fang, Jin; Yang, Zhenyu

VERSION 1 – REVIEW

REVIEWER	Lingling Gao School of Nursing, Sun Yat-sen University, Guangzhou, China
REVIEW RETURNED	22-Mar-2020

GENERAL COMMENTS	This paper presents interesting ideas and questions regarding the stated objective of this manuscript, to explore and compare predictors of exclusive breastfeeding (EBF) among migrant and non-migrant mothers in China. In order to improve the manuscript, I have some comments: 1. What the difference between the migrant and non-migrant mothers? Please explain in the Introduction section.2. In the results section, the differences on socio-demographic characteristics between the two groups may be analyzed.
--

REVIEWER	Bindi Borg University of Sydney, Australia
REVIEW RETURNED	12-Apr-2020

GENERAL COMMENTS	Predictors of exclusive breastfeeding practice among migrant and non-migrant mothers in China: results from a large-scale nationwide survey Overall comment: This study aimed to explore and compare predictors of exclusive breastfeeding (EBF) among migrant and non-migrant mothers in China. The study provides useful insights for programming and interventions that aim to improve exclusive breastfeeding rates. Introduction: 1. Lines 27: Does “unfair access to urban benefits” mean that migrants have more or less access? Clarify. 2. Lines 31-33: Expound briefly on the findings of “previous studies focused on immigrant mothers, who moved to another country where was different from their countries of origin” – what did those studies find? Methods:
--

	3. Line 47: psychiatric disorders but not other health factors as an exclusion criteria? Explain why. 4. What does it mean that “This research was done without involvement of patients and the public”? Were the respondents considered neither patients nor public? 5. “We defined infants who were fed exclusively with breastmilk in the past 24 hours as exclusive breastfeeding” – what about babies that received prelacteal feeds? Need to give some information on prelacteal feeding in China to explain or justify this exclusion. 6. Did the researchers test for collinearity between breastfeeding difficulties postpartum and low breastfeeding intentions within 1 month postpartum? It seems likely that there would be collinearity. Results: 7. Define early initiation of breastfeeding (WHO definition?). 8. How useful is breastfeeding intention after birth as a factor? Why was that chosen instead of breastfeeding intention before birth? Discussion: 9. Very interesting observations on supporting environments in large cities, and on key message of exclusive breastfeeding to 6 months. That is very useful for future programming and should be highlighted as important learnings of this study. 10. As in point 6, low intention and nipple pain suggest that low intention is collinear with problems breastfeeding. 11. What does “tertiary hospitals, where well-trained health workers and good service” signify? Less separation? More breastfeeding support? 12. In the section “Different from previous studies, we failed to find that breastfeeding difficulties were barriers to optimal breastfeeding practice”, the explanation for this finding is unclear. How and why did migrant mothers solve their breastfeeding difficulties? The authors talk about the experience elsewhere but not the experience of migrant mothers. 13. The section on EIBF needs a conclusion/recommendation, eg. EENC Care including EIBF should be implemented in order to improve breastfeeding rates. 14. It would be very interesting, not necessarily in this article, to look at the relationship between CS and failure to breastfeed amongst Chinese mothers. 15. The authors discuss the limitation about choice of hospitalization services which may be restricted for migrant mothers. Was data on hospital choice collected? Can that limitation be overcome? If it can be, then hospitalisation choice should be included as a variable. 16. It is worth discussing the implication of overestimation of the prevalence of exclusive breastfeeding i.e. that exclusive breastfeeding may be less than 30%, which is really problematic. Overall 17. As noted in points 6, 8, and 10, I doubt the utility of the “low breastfeeding intentions within 1 month postpartum” variable. General comment:  - Suggest using comma separator in numbers, eg. 10,408 for better readability - Suggest minor editing for spelling, grammar and readability.
--	---

VERSION 1 – AUTHOR RESPONSE

Reviewer(s)' Comments to Author:

Reviewer: 1

Reviewer Name

Lingling Gao

Institution and Country

School of Nursing, Sun Yat-sen University, Guangzhou, China

Please state any competing interests or state 'None declared':
None declared

Please leave your comments for the authors below

This paper presents interesting ideas and questions regarding the stated objective of this manuscript, to explore and compare predictors of exclusive breastfeeding (EBF) among migrant and non-migrant mothers in China. In order to improve the manuscript, I have some comments:

1. What the difference between the migrant and non-migrant mothers? Please explain in the Introduction section.

We added some explanations of the differences between migrant and non-migrant mothers in the first paragraph of the introduction section (Page 5, line 12-15).

In addition to socio-economic and cultural differences between migrant and non-migrant women in China, previous studies also found that migrant women had poor knowledge and limited utilization of maternal health care services.

2. In the results section, the differences on socio-demographic characteristics between the two groups may be analyzed.

In the last column of revised Table 1, we displayed P-values of Chi-square tests to compare differences in socio-demographic characteristics and other predictors of EBF between the two groups (Page 13, Table 1).

Migrant mothers were statistically different from non-migrant mothers in the following aspects such as age, ethnicity, occupation of themselves and their partners, formal employment status and place of residence (Page 10, Line 16-25).

Reviewer: 2

Reviewer Name

Bindi Borg

Institution and Country

University of Sydney, Australia

Please state any competing interests or state 'None declared':
None declared

Please leave your comments for the authors below
bmjopen-2020-038268 Review

Predictors of exclusive breastfeeding practice among migrant and non-migrant mothers in China:
results from a large-scale nationwide survey

Overall comment:

This study aimed to explore and compare predictors of exclusive breastfeeding (EBF) among migrant and non-migrant mothers in China. The study provides useful insights for programming and interventions that aim to improve exclusive breastfeeding rates.

Introduction:

1. Lines 27: Does “unfair access to urban benefits” mean that migrants have more or less access? Clarify.

We revised “unfair access to urban benefits” to “less potential access to urban benefits” to clarify it (Page 5, line 11).

2. Lines 31-33: Expound briefly on the findings of “previous studies focused on immigrant mothers, who moved to another country where was different from their countries of origin” – what did those studies find?

We added further explanation on findings of previous studies in the second paragraph of the introduction section (Page 5, line 17-19).

Previous study found that immigrant women were more likely to initiate and continue breastfeeding than non-immigrant. But exclusive breastfeeding remains a challenge for both groups.

Methods:

3. Line 47: psychiatric disorders but not other health factors as an exclusion criteria? Explain why.

Our main variable of interest is the prevalence of exclusive breastfeeding, which is based on maternal recall. In addition, we tried to cover as many target populations as possible. In order to make sure that the mothers can answer questions correctly and clearly, we used psychiatric disorders as exclusion criteria.

4. What does it mean that “This research was done without involvement of patients and the public”? Were the respondents considered neither patients nor public?

“Patient and Public involvement” was required by the journal, and the main objective of this is to support co-production of research. We revised this part to clarify the involvement of the patient and the public in our study (Page 7, Line 5-9).

Neither these sampled mothers nor the public were involved in the study design and implementation. The study results will be disseminated to the public through media briefings and scientific publications.

5. “We defined infants who were fed exclusively with breastmilk in the past 24 hours as exclusive breastfeeding” – what about babies that received prelacteal feeds? Need to give some information on prelacteal feeding in China to explain or justify this exclusion.

Previous studies indicate that providing prelacteal feeds is a long-held tradition in many parts of China, but regional disparities are quite large. Water is the top one drinks received by the newborn babies in China. [1]

The reason why we used the WHO 24 hour recall of infant feeding in our study is: 1) to generate internationally comparable measures of exclusive breastfeeding; 2) to avoid recall bias of mothers with older infants.

6. Did the researchers test for collinearity between breastfeeding difficulties postpartum and low breastfeeding intentions within 1 month postpartum? It seems likely that there would be collinearity.

The correlation between breastfeeding difficulties postpartum and low breastfeeding intentions within 1 month postpartum is 0.1546, which is quite low. In our questionnaire, we asked the respondents to list reasons for low breastfeeding intention within 1 month postpartum. Cracked nipples or nipple pain, insufficient breastmilk supply ranked as the top 2 reasons for low

breastfeeding intentions, which accounted for 30.4% and 18.1% percent of all answers respectively. Thus, we treat these two variables separately.

Results:

7. Define early initiation of breastfeeding (WHO definition?).

Yes, we used the definition of the WHO to define early initiation of breastfeeding. We defined it as the practice that infants were put to the breast of their mothers within the first hour of birth (Page 8, line 17-19).

8. How useful is breastfeeding intention after birth as a factor? Why was that chosen instead of breastfeeding intention before birth?

To be honest, we didn't design questions about breastfeeding intention before birth in the questionnaire. We asked the mothers when they started to think about how to feed their babies instead (pre-pregnancy, during pregnancy and after birth). We added this variable into statistical analysis. As what can be confirmed in the revised Table 1 that non-migrant mothers think about how to feed their babies earlier than migrant mothers in general ($P < 0.001$). While, it was only significantly associated with the prevalence of exclusive breastfeeding among migrant mothers in univariate analysis (Page 13, Table 1).

In addition, we have a well-documented custom which is known as "sitting the month" or postpartum confinement in the first month postpartum in China. During this time, new mothers are confined to their homes, where they need to follow some traditions and rules to control their diet and physical activity. From our observation and own experience, the first month postpartum is a very critical period which will influence the mental health of mothers and their breastfeeding outcomes afterwards. Thus, we included breastfeeding intention within 1 month postpartum as a predictor considering this cultural context.

Discussion:

9. Very interesting observations on supporting environments in large cities, and on key message of exclusive breastfeeding to 6 months. That is very useful for future programming and should be highlighted as important learnings of this study.

Thank you very much, we added it in the important learnings of this study (page 4, line 8-9).

10. As in point 6, low intention and nipple pain suggest that low intention is collinear with problems breastfeeding.

As we explained earlier that even though breastfeeding difficulties is the largest reason for low breastfeeding intention in the first month, it only accounted for 30% of the answers.

11. What does “tertiary hospitals, where well-trained health workers and good service” signify? Less separation? More breastfeeding support?

Hospitals in China are usually classified into a 3-tier system and designated as Primary, Secondary, or Tertiary institutions. Tertiary hospitals usually have more clinical expertise and experience and more likely to be baby-friendly hospitals.

There are limited studies on breastfeeding support of NICU healthcare professions in China. One previous study found that knowledge about breastfeeding preterm infants among NICU healthcare professions in China was limited using data from 9 tertiary NICUs, and they inferred that knowledge deficit in the smaller hospitals may be worse [2]. While there is no comprehensive study at present to support this argument.

Among our sampled mothers with premature infants, the share of migrant mothers who delivered in hospitals at municipal or higher levels of 65.9% was lower than non-migrant mothers of 73.6%. Internationally wide-known practices such as kangaroo care (KC) for promoting the development of premature infants have only been implemented in some NICUs of high-level hospitals in China as pilot studies (page 21, line 2-11).

Thus, we think tertiary hospitals may encourage more involvement of parents to care their infant through practice like KC.

12. In the section “Different from previous studies, we failed to find that breastfeeding difficulties were barriers to optimal breastfeeding practice”, the explanation for this finding is unclear. How and why did migrant mothers solve their breastfeeding difficulties? The authors talk about the experience elsewhere but not the experience of migrant mothers.

After we controlling for additional confounding factors (maternal and paternal occupation, delivery hospital level etc.), we failed to find that breastfeeding difficulties were a significant predictor of

exclusive breastfeeding anymore in the multiple logistic regressions. Thus, we excluded this part from the discussion section of our manuscript.

Our data suggested that mothers looked for help from multiple sources. Family members, health professionals, and themselves ranked as the top three ones to which they looked for help. Because it is a multiple-choice question, we couldn't differentiate who is most helpful when they encountered difficulties. In addition to looking for help from different people, around 50% of the mothers also looked for help through the internet, which could overcome the breastfeeding difficulty issue.

13. The section on EIBF needs a conclusion/recommendation, eg. EENC Care including EIBF should be implemented in order to improve breastfeeding rates.

Thank you very much for your suggestion. We added two recommendations. The first one is that hospitals in China need to speed up their process of adopting EENC recommended practices. Secondly, China needs to establish national policies to help hospitals and health workers to overcome obstacles in implementing these practices. (page 22, line 4-6)

14. It would be very interesting, not necessarily in this article, to look at the relationship between CS and failure to breastfeed amongst Chinese mothers.

Thank you very much for your advice, we will consider it as the next research topic. For your reference, in our sample, the rate of cesarean is quite high, which is around 38.1%. Among mothers who delivered their baby by CS, 96.6% of them had ever practiced breastfeeding, while only 25.7% of them practiced exclusive breastfeeding.

15. The authors discuss the limitation about choice of hospitalization services which may be restricted for migrant mothers. Was data on hospital choice collected? Can that limitation be overcome? If it can be, then hospitalisation choice should be included as a variable.

We only collected data on the hospital level where mothers delivered their infants (hospitals at municipal level or above, hospitals at county level, and others) and included it as a predictor of EBF. We found that more migrant mothers delivered in hospitals at municipal level or above, and hospital levels have no effect on the practice of exclusive breastfeeding (Page 13, Table 1).

In general, hospitals at a higher level are considerably better in general. Due to the lack of social networks or information, it may be difficult for migrant mothers to get into hospitals with better

obstetrical departments. Because of limited information, we couldn't provide further evidence to support this assumption. We will leave it for future research.

16. It is worth discussing the implication of overestimation of the prevalence of exclusive breastfeeding i.e. that exclusive breastfeeding may be less than 30%, which is really problematic.

Thank you very much for this advice. We included some discussion on the cost of low breastfeeding in terms of early childhood development and maternal health (page 23, line 8-10).

Overall

17. As noted in points 6, 8, and 10, I doubt the utility of the "low breastfeeding intentions within 1 month postpartum" variable.

As we explained earlier, low breastfeeding intentions within 1 month postpartum and breastfeeding difficulties indicate different things in this study. We included this variable mainly considered the special cultural background in China.

In this revised version, we included more discussions on this issue (page 20, line 6-18).

General comment:

- Suggest using comma separator in numbers, eg. 10,408 for better readability

Thank you very much for this advice, we revised it accordingly.

- Suggest minor editing for spelling, grammar and readability.

Thank you very much for this advice, we asked an English-native speaker to proofread it.

Reference

- [1] Tang L, Hewitt K, Yu C. Prolactin Feeds in China. *Curr Pediatr Rev* 2012;8:304–12. <https://doi.org/10.2174/157339612803307688>.
- [2] Yang Y, Li R, Wang J, Huang Q, Lu H. Knowledge of healthcare providers regarding breastfeeding preterm infants in mainland China. *BMC Pediatr* 2018;18:251. <https://doi.org/10.1186/s12887-018-1223-7>.

VERSION 2 – REVIEW

REVIEWER	Lingling Gao School of Nursing, Sun Yat-sen University, Guangzhou, China
REVIEW RETURNED	11-Jun-2020

GENERAL COMMENTS	This study has the potential to provide important insights on exclusive breastfeeding in migrant and non-migrant mothers in urban China. However, I have concern regarding the literature review and methodology. The literature review in Introduction section is insufficient. The risk factors of exclusive breastfeeding have not been reviewed and summarized. I also have concerns on the questionnaire used in this study. How is it developed? Moreover, more information is needed on the number of items, scale of response and psychometric properties of the questionnaire. Please provide examples of the items.
---

REVIEWER	Bindi Borg University of Sydney, Australia
REVIEW RETURNED	07-Jun-2020

GENERAL COMMENTS	Most of my questions/suggestions in the review were satisfied. However, I am still concerned about two points (numbers refer to earlier review): Methods: 5. “We defined infants who were fed exclusively with breastmilk in the past 24 hours as exclusive breastfeeding” – what about babies that received prelacteal feeds? Need to give some information on prelacteal feeding in China to explain or justify this exclusion – can use the author’s response to explain prelacteal feeding practices. 6. Did the researchers test for collinearity between breastfeeding difficulties postpartum and low breastfeeding intentions within 1 month postpartum? It seems likely that there would be collinearity. Need to define breastfeeding intentions in the article, as they have in the author’s response. Usually, breastfeeding intentions means something a little different to this.
--

VERSION 2 – AUTHOR RESPONSE

Reviewer(s)' Comments to Author:

Reviewer: 2

Reviewer Name

Bindi Borg

Institution and Country

University of Sydney, Australia

Please state any competing interests or state 'None declared':

None declared

Please leave your comments for the authors below

Most of my questions/suggestions in the review were satisfied. However, I am still concerned about two points (numbers refer to earlier review):

Methods:

5. “We defined infants who were fed exclusively with breastmilk in the past 24 hours as exclusive breastfeeding” – what about babies that received prelacteal feeds? Need to give some information on prelacteal feeding in China to explain or justify this exclusion – can use the author’s response to explain prelacteal feeding practices.

Thank you very much for this advice. In the revised version of the manuscript, we added further explanation about why we used the WHO 24-hour recall method in our study and excluded prelacteal feeds in our analysis (page 8, line 1-10).

6. Did the researchers test for collinearity between breastfeeding difficulties postpartum and low breastfeeding intentions within 1 month postpartum? It seems likely that there would be collinearity. Need to define breastfeeding intentions in the article, as they have in the author’s response. Usually, breastfeeding intentions means something a little different to this.

We are sorry that we didn’t answer this question clearly. For the breastfeeding intention, we asked the mothers “How often were you unwilling to breastfeed in the first month postpartum”, and we defined mother who answered “always, very often or sometimes” as having a low intention of breastfeeding within 1 month postpartum. And then we further asked the mothers who answered “always, very often or sometimes” to list reasons for low breastfeeding intention within 1 month postpartum.

We used the STATA command *collin* to test the collinearity between breastfeeding difficulties postpartum and low breastfeeding intentions within 1 month postpartum. The mean variance inflation factors (VIF) of the two variables are 1.02 which is far less than 10. Thus, we think it is reasonable to treat these two variables differently.

We added a detailed definition of this variable in the revised manuscript to clarify its meaning (page 9 line 5-7).

Reviewer: 1

Reviewer Name

Lingling Gao

Institution and Country

School of Nursing, Sun Yat-sen University, Guangzhou, China

Please state any competing interests or state 'None declared':

No

Please leave your comments for the authors below

This study has the potential to provide important insights on exclusive breastfeeding in migrant and non-migrant mothers in urban China. However, I have concern regarding the literature review and methodology. The literature review in Introduction section is insufficient. The risk factors of exclusive breastfeeding have not been reviewed and summarized. I also have concerns on the questionnaire used in this study. How is it developed? Moreover, more information is needed on the number of items, scale of response and psychometric properties of the questionnaire. Please provide examples of the items.

Thank you very much for this advice. We reviewed and summarized risk factors of exclusive breastfeeding in the revised introduction part (page 5, line 22-page 6 line 3).

For the question on how the questionnaire was developed, the first draft of it was mainly developed by the authors from the National Institute for Nutrition and Health, Chinese Center for Disease Control and Prevention (NINH, China CDC) who is in charge of the Chinese National Nutrition and Health Survey (CNNHS) of China. Thus, questionnaire used in this survey was partly based on CNNHS and further added influencing factors of breastfeeding practice. We largely referred to the systematic review in the *Lancet Breastfeeding Series 2016* to design questions on the influencing factors of breastfeeding and at the same time considered the cultural background of China [1].

Two rounds of the consultative meetings were held to refine the questionnaire with experts specialized in sampling, public health, epidemiology, breastfeeding counseling, and sociology. The questionnaire was tested in Beijing with 30 participants in August 2017. Based on the issues found in this pilot survey, a meeting was arranged to further refine and finalize the questionnaire. A total number of 130 items were finally designed in the questionnaire.

We didn't use the psychological scaling method to measure and define variables, thus we couldn't provide psychometric property of the questionnaire.

For the scale of response, we used dichotomous styled Yes or No questions, the Likert scale questions, nominal questions, and ordinal questions.

For example, for the question of "Is your baby a premature birth", we used the dichotomous styled Yes or No as the answer.

For the question of "how often you were unwilling to breastfeed in the first month postpartum", we used a 5-point Likert scale to measure the frequency (always, very often, sometimes, rarely, and never).

For the question of “education level of the mother”, we used the ordinal answers which ranged from 1 for no formal education to 8 for graduate school or above.

For the occupation category of the mothers, we used the nominal answers which consisted of 11 categories such as agriculture, forestry, animal husbandry, fishery and water conservancy practitioner, production and transportation equipment operators and related personnel, business and service industry practitioner, etc.

Reference

- 1 Rollins NC, Bhandari N, Hajeebhoy N, *et al.* Why invest, and what it will take to improve breastfeeding practices? *Lancet* 2016;**387**:491–504. doi:10.1016/S0140-6736(15)01044-2

VERSION 3 – REVIEW

REVIEWER	Lingling Gao School of Nursing, Sun Yat-sen University, Guangzhou, China
REVIEW RETURNED	03-Jul-2020

GENERAL COMMENTS	The manuscript has been improved after this second review. However, I still have several concerns regarding the methodology. How about the validity and reliability of the questionnaire? Moreover, I have concerns on the supportive influences measures (are we sure that one item scale with a “yes” “no” response format is enough to define a condition?). What is Even well-established single items are used, usually the response is provided in a 5 or 7 Likert scale. How was this information collected? When and who was present when this measure was administered? I would expect that the risk of socially desirable responding to this type of question and answer format would be very high.
---

REVIEWER	Bindi Borg University of Sydney Australia
REVIEW RETURNED	06-Jul-2020

GENERAL COMMENTS	All my comments on the previous versions have now been addressed.
---

VERSION 3 – AUTHOR RESPONSE

Reviewer(s)' Comments to Author:

Reviewer: 1

Reviewer Name

Lingling Gao

Institution and Country

School of Nursing, Sun Yat-sen University, Guangzhou, China

Please state any competing interests or state 'None declared':

None declared

Please leave your comments for the authors below

The manuscript has been improved after this second review. However, I still have several concerns regarding the methodology. How about the validity and reliability of the questionnaire?

Thank you very much for raising concerns regarding the methodology. Since the core items of infant feeding indicators in the questionnaire are from the World Health Organization standard infant and young child feeding indicator questionnaire, validity and reliability are not tested again[1,2].

Moreover, I have concerns on the supportive influences measures (are we sure that one item scale with a "yes" "no" response format is enough to define a condition?).

For covariate variables used in this study, we used "yes" or "no" response format for the original questions of the following 5 variables: 1) Premature birth (Is the baby a premature birth?); 2) Breastfeeding difficulties postpartum (After discharge from the hospital, have you experienced symptoms such as cracked nipples, abnormal nipples, breast tenderness, blocked breast ducts, and mastitis?); 3) attend antenatal visits (Have you participated in the antenatal visits); 4) attend breastfeeding education sessions (Have you participated breastfeeding education sessions at the healthcare facility during pregnancy?); 5) attend mother groups (Have you ever participated in mother groups such as Wechat group, App mother group, etc.).

For other covariate variables, we used fill-in format questions, multiple-choice questions with forced-choice or Likert scales.

Even well-established single items are used, usually the response is provided in a 5 or 7 Likert scale. How was this information collected? When and who was present when this measure was administered? I would expect that the risk of socially desirable responding to this type of question and answer format would be very high.

The information was collected through face-to-face interviews. To decrease the reporting bias, we only invited mothers to participate in the study. Additionally, considering that the crying of infants after vaccination may disturb interviews, we invited mothers in the waiting areas of the immunization clinics

before vaccination. During the process of data collection, we set up a relatively independent space for the enumerators to collect data at the immunization clinic. If the mother came to the immunization clinics with her infants only, infants were always present when this measure was administered. If the mother came with other family members, we asked the mothers to participate in the interview without the presence of her family members.

For subjective questions such as “how often you were unwilling to breastfeed in the first month postpartum”, we asked the enumerators to read out all the options of answers and let the mothers choose by themselves. At the same time, we also asked the enumerators not to conduct suggestive questioning. Even though we tried to reduce social desirability bias through questioning techniques, we cannot rule out the possibility that social desirability bias did exist. Thus, in the discussion part, we added some information about social desirability bias as the limitation of this study (Page 23, line 15-17).

Reviewer: 2

Reviewer Name

Bindi Borg

Institution and Country

University of Sydney
Australia

Please state any competing interests or state 'None declared':

None declared

Please leave your comments for the authors below

All my comments on the previous versions have now been addressed.

Thank you very much for your comments.

Reference

- 1 World Health Organization. Indicators for assessing infant and young child feeding practices: part 2: measurement. 2010.
- 2 Li R, Scanlon KS, Serdula MK. The validity and reliability of maternal recall of breastfeeding practice. *Nutr. Rev.* 2005. doi:10.1301/nr.2005.apr.103-110

VERSION 4 – REVIEW

REVIEWER	Lingling Gao School of Nursing, Sun Yat-sen University, Guangzhou, China
REVIEW RETURNED	20-Jul-2020
GENERAL COMMENTS	What I have commented have been improved. I have no more suggestions.